# Current Therapy and Liver Transplantation for Portopulmonary Hypertension in Japan

**DOI:** 10.3390/jcm12020562

**Published:** 2023-01-10

**Authors:** Katsutoshi Tokushige, Tomomi Kogiso, Hiroto Egawa

**Affiliations:** 1Department of Internal Medicine and Gastroenterology, Tokyo Women’s Medical University, 8-1 Kawada-cho, Shinjuku-ku, Tokyo 162-8666, Japan; 2Department of Hepatopancreatic Surgery, Tokyo Women’s Medical University, 8-1 Kawada-cho, Shinjuku-ku, Tokyo 162-8666, Japan

**Keywords:** portopulmonary hypertension, endothelin receptor antagonists, liver transplantation

## Abstract

Portopulmonary hypertension (PoPH) and hepatopulmonary syndrome are severe pulmonary complications associated with liver cirrhosis (LC) and portal hypertension. Three key pathways, involving endothelin, nitric oxide, and prostacyclin, have been identified in the development and progression of pulmonary arterial hypertension (PAH). To obtain a good effect with PAH-specific drugs in PoPH patients, it is important to diagnose PoPH at an early stage and promptly initiate therapy. The majority of therapeutic drugs are contraindicated for Child-Pugh grade C LC, and their effects decrease in the severe PAH stage. Among many LC patients, the measurement of serum brain natriuretic peptide levels might be useful for detecting PoPH. Previously, liver transplantation (LT) for PoPH was contraindicated; however, the indications for LT are changing and now take into account how well the PoPH is controlled by therapeutic drugs. In Japan, new registration criteria for deceased-donor LT have been established for PoPH patients. PoPH patients with a mean pulmonary arterial pressure <35 mmHg and pulmonary vascular resistance <400 dyn/s/cm^−5^ are indicated for LT, regardless of whether they are using therapeutic drugs. Combined with PAH-specific drugs, LT may lead to excellent long-term outcomes in PoPH patients. We aimed to review current therapies for PoPH, including LT.

## 1. Introduction

The two major pulmonary vascular consequences of portal hypertension are hepatopulmonary syndrome (HPS) and portopulmonary hypertension (PoPH) [1]. PoPH is defined as pulmonary arterial hypertension (PAH) associated with intrahepatic or extrahepatic portal hypertension. McDonnell et al. [2,3] showed a prevalence of histopathological changes of 0.61% in autopsies of patients with liver cirrhosis (LC), and PoPH is the fourth most common condition (approximately 10%) among PAH cases [2]. Recent cohort studies showed prevalence rates of PoPH of 1–6% in patients presenting to be evaluated for liver transplantation (LT) [3,4,5,6]. Patients with PoPH have an increased risk of death; their reported death rates without therapy at 1 and 5 years were 54% and 86%, respectively [7]. In many cases, PoPH greatly complicates or precludes LT and thus significantly affects the course of hepatic failure in these patients [8,9].

PAH, including PoPH, is often considered to be driven by vasoconstriction and sclerosis of the pulmonary arteries, and our case showed the typical lung pathological changes that accompany PoPH (Figure 1). Three key pathways, involving endothelin, nitric oxide, and prostacyclin, have been identified in the development and progression of PAH. PAH-specific therapeutic approaches concentrate on these characteristics, with drugs targeting endothelin receptors (e.g., macitentan, bosentan), phosphodiesterase-5 (e.g., sildenafil), or the prostacyclin receptor (e.g., beraprost) [10,11]. These drugs dramatically improve the condition and survival rates of patients with PAH. However, the improvement in long-term survival brought about by these drugs in patients with PoPH is not yet clear. In addition, while LT was considered a contraindication for PoPH, the indications for LT in patients with PoPH are changing owing to the development of new therapeutic drugs. Thus, we conducted the present study to reassess the current therapies for PoPH, including LT.

## 2. Clinical Features of PoPH

In Japan, there are approximately 0.4–0.5 million patients with LC [12]. Atsukawa et al. reported that the prevalence of PoPH was approximately 1% (2/186) in Japanese patients with LC. In addition, the presence of PoPH and high pulmonary artery pressure were not associated with the degree of hepatic functional reserve or hepatic vein pressure gradient [13]. The most common pulmonary symptoms include fatigue, exertional dyspnea, syncope, and chest pain [1]. However, it is difficult to diagnose PoPH based on symptoms because patients with LC frequently experience general fatigue and dyspnea. Actually, many factors, including the presence of pleural effusion, may result in general malaise and dyspnea, such as anemia, hepatopulmonary syndrome, and hepatorenal syndrome, among others. In patients with PoPH, there may also be evidence of right ventricular failure. Chest radiographs may show cardiomegaly and prominent pulmonary arteries with peripheral vessel pruning. An electrocardiogram (ECG) may demonstrate right ventricular hypertrophy, right axis deviation, and right bundle branch block, which are features of right heart strain [1]. Echocardiography should be performed when PoPH is suspected based on these clinical examinations and findings. If the right ventricular systolic pressure (RVSP) is ≥50 mmHg, PoPH is highly suspected [5]. In such cases, a cardiologist is consulted and right heart catheterization (RHC) should be performed.

In our opinion, it is ideal to perform echocardiography once in all patients with LC. However, this treatment option has certain problems in terms of cost and manpower given the low prevalence of PoPH. Echocardiography should be performed in selected LC patients, considering pulmonary symptoms, background liver diseases, and serum BNP.

To obtain a definitive diagnosis of PoPH, PAH must be determined based on a specific hemodynamic profile. This profile includes a resting mean pulmonary arterial pressure (mPAP) ≥25 mmHg, pulmonary artery wedge pressure (PAWP) ≤15 mmHg, and pulmonary vascular resistance (PVR) >240 dyn/s/cm^−5^ (3 Wood units) [1,5]. In the 6th World Symposium on Pulmonary Hypertension, convincing epidemiological data have provided a rationale that justifies a revision in the hemodynamic definition of PH as an mPAP of > 20 mmHg [14].Portal hypertension is defined as a portal venous gradient of > 5 mmHg. In general, if LC or portal hypertension is diagnosed by liver biopsy or imaging, the measurement of portal vein pressure is not necessary.

In our department, from 2000 to 2020, we encountered seven patients with PoPH and seven patients with HPS (Table 1 and Table 2) [12]. These two sets of patients were quite different in terms of their sex and background liver diseases, but the age at diagnosis was almost similar. All seven patients with PoPH were female (100%) (female <%>, 100% in PoPH vs. 29% in HPS, *p* < 0.01) and five had non-viral liver diseases (two with primary biliary cholangitis <PBC>, one with nonalcoholic steatohepatitis <NASH> and two with portal vein obstruction). Of the other two patients, one had PBC with anti-mitochondrial antibodies and hepatitis C virus (HCV)-RNA, and the other had LC due to HCV. Of the seven patients with HPS, two (29%) were female, four (57%) had LC due to HCV, two had NASH with hypopituitarism, and one had alcoholic cirrhosis. In Japan, LC with HCV is common in 40–50% [12]. Kawut et al. reported that female sex and the presence of autoimmune liver disease were associated with an increased risk of PoPH, whereas hepatitis C infection was associated with a decreased risk of PoPH in patients with advanced liver disease [15]. Atsukawa et al. [13] reported high PAP values in patients with PBC or autoimmune hepatitis. Therefore, hormonal and immunological factors may be integral to PoPH [13,15,16]. In addition, all seven patients with PoPH had huge portosystemic shunts (three cases) or esophageal varices with surgical or endoscopic therapies. When physicians examine cases of LC in females and LC with portosystemic shunt, varices, or PBC, they should consider the possibility of PoPH.

Among the 14 patients with PoPH or HPS, three cases of NASH (21%) induced by hypopituitarism, which is very rare among LC cases, were observed. Several papers have reported LC cases with hypopituitarism that were complicated with PoPH or HPS [17,18,19]. In addition, the insulin-like growth factor is associated with PAH [20,21]. Therefore, when physicians examine patients with LC with hypopituitarism, they should be aware of the possible presence of PoPH or HPS.

Recently, lenvatinib and sorafenib have been frequently used as therapies for hepatocellular carcinoma (HCC) [22,23]. These are vascular endothelial growth factor (VEGF) targeting and/or tyrosine kinase inhibitor (TKI) drugs, and increase the levels of the vasoconstrictor endothelin, which binds to receptors on endothelial cells, causing smooth muscle contraction and increased vessel resistance and blood pressure. Ishikawa et al. [24] published a case report of PoPH exacerbated by the administration of lenvatinib for HCC, suggesting that these drugs might worsen potential PoPH. When administering lenvatinib or sorafenib for HCC, the possibility of potential PoPH should be considered.

Serum brain natriuretic peptide (BNP) may be a candidate serum diagnostic biomarker of PoPH. In our study, RVSP and BNP levels were weakly correlated (r = 0.40, *p* = 0.01). Yoshimaru et al. [25] reported that patients with PoPH had a significantly higher BNP level, which was predictive of asymptomatic PoPH, with an optimal cut-off value of 29.1 pg/mL. In our data, it was 90.8 pg/mL in patients with RVSP ≥ 36 mmHg [6]. Out of seven HPS patients, we measured serum BNP in five patients. In our data, BNP in HPSs was slightly increased, but the mean BNP of PoPH was significantly higher than that of HPS (mean serum BNP; 40.6 + 17.2 in HPS; 169 + 250.3 in PoPH < 0.05). The establishing of a serum diagnostic biomarker for PoPH should be the goal of future research.

If the reported percentage of PoPH in patients with LC of 1–6% is correct, the number of patients with PoPH in our department should be more than 7 over 20 years, and we might have missed some cases. We must carefully consider the possible presence of PoPH when examining patients with LC or portal hypertension.

## 3. Drug Therapy for PoPH

PAH-specific therapeutic approaches concentrate on characteristics with drugs targeting endothelin receptors, phosphodiesterase-5, or the prostacyclin receptor [10,11]. In particular, combination therapy with these drugs dramatically improves the survival rate of patients with PAH [26]. However, the long-term effects of these drugs on PoPH and the prognosis of patients receiving these therapies remain unclear.

There are two main challenges to the provision of drug therapy for PoPH. First, the prognosis for patients with PoPH is poorer than that for patients with conventional PAH, especially since the lifespan of patients with LC is limited. When considering the prognosis of LC, the clinical benefit of therapy for PoPH is small. Second, determining the appropriate use of drugs for PAH is difficult in patients with LC. Because the serum concentration of drugs can frequently increase in patients with LC, side effects can be easily induced.

Recently, the results of several drug trials involving patients with PoPH have been reported. Hoeper et al. [27] reported trials of inhaled iloprost and bosentan (endothelin receptor antagonists) in patients with PoPH. Patients with PoPH treated with bosentan had higher survival rates, and the therapy was relatively safe. Sitbon et al. [28] reported that 85 patients with PoPH were randomly assigned to receive macitentan (n = 43) or a placebo (n = 42). To our knowledge, this is the first prospective, randomized drug trial involving PoPH. Macitentan significantly improved PVR in patients with PoPH with no hepatic safety concerns. Recently, Savale et al. [29] conducted a retrospective study of 637 patients with PoPH. Most patients initially received monotherapy, either with a phosphodiesterase-5 inhibitor (n = 336) or an endothelin receptor antagonist (n = 128); 95 (15%) initially received double oral combination therapy. After the median treatment, there were significant improvements in the functional class, 6 min walk distance (6MWD), and PVR. Combination therapy has shown better results than monotherapy. Takahashi et al. [30] reported the clinical features and effects of PAH-specific drugs in Japanese patients with PoPH. Combined therapy showed good effect, and Japanese patients with PoPH showed higher cardiac outputs (COs) and cardiac indexes (CIs), better exercise tolerance, and lower PVRs than patients with idiopathic/heritable PAH.

In our department, we encountered seven patients, five of whom received drug therapies. Combination therapy and endothelin receptor antagonist monotherapy were effective (Figure 2). The clinical course of the most effective case (Case #2) is shown in Figure 3. After the therapy, serum BNP, mPAP, PVR, and TGF-β, which are among the most important factors for hepatic fibrosis, decreased, and the 6MWD improved. In addition, improved data and conditions were maintained for 12 years. A case in which the therapy was ineffective was of severe PH, with Child-Pugh grade C. Therefore, early diagnosis and introduction of medical therapy for PAH are important.

Peripheral and portal vein serum endothelin concentrations are markedly increased in patients with LC [31]. The use of endothelin antagonism in portal hypertensive mice decreases the portal vein pressure, TGF-β production, and portal fibrosis area [32,33]. These findings suggest that endothelin receptor antagonists, such as bosentan and macitentan, might have beneficial effects on liver fibrosis and portal hypertension in humans and may be a key medication among the therapeutic drugs for PoPH. Bosentan frequently induces liver dysfunction and should not be used to treat patients with severe LC. However, macitentan has demonstrated long-term efficacy in PAH, with a good hepatic safety profile, even in patients with LC, excluding those with Child-Pugh grade C disease. Most PAH drugs are contraindicated for LC with Child-Pugh grade C, suggesting that it is important to diagnose PoPH at an early stage.

## 4. LT with PoPH

PoPH is associated with significant intraoperative and postoperative morbidity and mortality. When LT has been performed in PoPH cases with an mPAP of ≥50 mmHg, the cardiopulmonary mortality was 100% [34], and moderate-to-severe PoPH has traditionally been considered a contraindication to LT. Savale et al. reported that the survival of patients with PoPH is strongly associated with the severity of liver disease, and patients who underwent LT showed the best long-term outcomes [29]. Therefore, new criteria and indications for LT are required for patients with PoPH.

A retrospective study of 1205 patients who underwent LT revealed no additional mortality if the systolic PAP was <60 mmHg [35]. A higher value resulted in a mortality rate of 42% nine months after the procedure and an overall poor quality of life. The three-year survival after LT was reduced in patients with moderate-to-severe PAH compared with that in patients with mild or no PAH [35]. In contrast, it was reported that in cases with an mPAP ≤ 35 mmHg, mortality was not affected [34,36] compared with that in patients with LC without PAH. In patients with an mPAP of 35 to <50 mmHg and a PVR of ≥250 dynes/s/cm, the mortality rate was 50% [34]. Patients with an mPAP of ≤35 mmHg are considered to be at low risk for mortality, while in patients with an mPAP of 35 to <50 mmHg, mortality is dependent on PVR and right heart function. An mPAP of ≥50 mmHg is a contraindication for LT [34,36,37].

In the USA, a recent case series using newer medical regimens reported favorable short-term LT outcomes in patients with moderate PoPH who achieved an mPAP <35 mmHg in response to medical therapy as long as the patient’s PVR was <400 dyn/s/cm^−5^ prior to surgery [7,8,36]. The current UNOS policy allows an MELD exception in patients with PoPH with an MELD score of 22 with baseline mPAP > 35 mmHg, provided there is documentation of a post-vasodilator treatment response to RHC with mPAP < 35 mmHg, PVR < 400 dyn/s/cm^−5^, and normal right ventricular function. The MELD score may increase every 3 months only if RHC shows that both mPAP and PVR remain within the target ranges [37].

In Japan, new registration criteria for deceased-donor liver transplantation (DDLT) have been established for patients [38]. The principal indication for DDLT is chronic liver failure in Japan; patients with a Child-Pugh score of ≥10 (Child–Pugh grade of C) would be registered. However, in cases with PoPH, if the mPAP is > 35 mmHg or the PVR is >400 dyn/s/cm^−5^ before treatment, and treatment decreases the mPAP to ≤ 35 mmHg and the PVR to ≤400 dyn/s/cm^−5^, even a patient with a Child-Pugh grade of A or B could register for transplantation. During the follow-up period, it should be confirmed every 3 months that the mPAP is ≤35 mmHg. The criteria for living-donor LT in patients with PoPH have not been determined. It is thought that patients with PoPH with an mPAP of ≤35 mmHg and a Child-Pugh grade of A or B are indicated for LT, regardless of whether they are using therapeutic drugs. Based on the above, we propose a flowchart to determine whether LT is indicated in PoPH cases (Figure 4). In patients with an mPAP of 35 to <50 mmHg, the possibility of LT is dependent on PVR and right heart function. In cases with mPA >35 mmHg and PVR >400 dyn/s/cm^−5^, LT is contraindicated. If LC progresses to a Child-Pugh grade of C, it is difficult to manage mPAP with therapeutic drugs. Most PAH-specific drugs are contraindicated for Child-Pugh grade C.

After LT, a significant decrease in PAP has been observed in patients with PoPH [28]. Among survivors of LT, PAH therapy was simplified from combination therapy to monotherapy in 16% of patients and discontinued in 22% of patients [28]. Overall, these findings show that if the mean PA can be controlled by drug therapy, LT should be considered to achieve excellent long-term survival.

## 5. Conclusions

It is important to proactively suspect PoPH and diagnose it as early as possible when we examine patients with LC. This can be done based on a medical interview, ECG, chest radiography, patient medical history examination for the presence of background liver diseases, and echocardiography. Although PoPH is a severe complication, excellent long-term survival can be expected if the condition is treated with a combination of therapeutic drugs and LT.

## Figures and Tables

**Figure 1 jcm-12-00562-f001:**
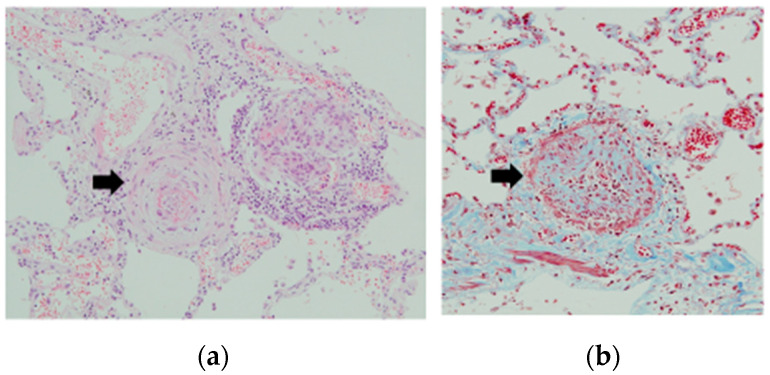
Autopsy of PoPH patient #1. Hematoxylin-eosin stain (**a**) and Masson trichrome staining (**b**). Typical histology of pulmonary arteries (PA). Severe sclerosis of pulmonary arteries. (**a**) Hypertrophy of the internal membrane of the small pulmonary arteries and narrow lumen, compatible with severe pulmonary arterial hypertension. (**b**) Proliferation of the elastic fibers was found in the pulmonary arteries using Masson trichrome staining.

**Figure 2 jcm-12-00562-f002:**
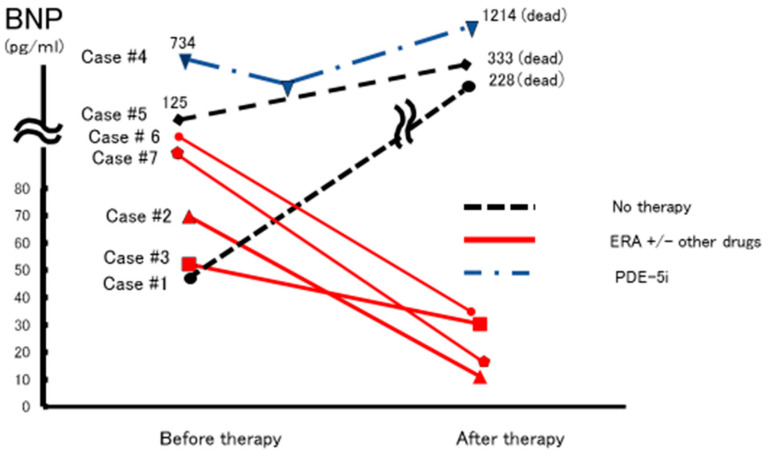
Changes in serum BNP levels in seven patients with PoPH. In patients #2, #3, #6, and #7, the serum BNP levels were decreased after therapy. Combination therapy (ERA and PDE5i or PRA) in patients #2, #6, and #7 and ERA in patient #3 were effective. In patient #4, PDE-5i monotherapy was ineffective and the patient died. In patients with PoPH who did not receive therapeutic drugs (Cases #1 and #6), serum BNP levels increased and the patients died. BNP, brain natriuretic peptide; ERA, endothelin receptor antagonists; PDE-5i, phosphodiesterase-5 inhibitor; PRA, prostacyclin receptor agonist.

**Figure 3 jcm-12-00562-f003:**
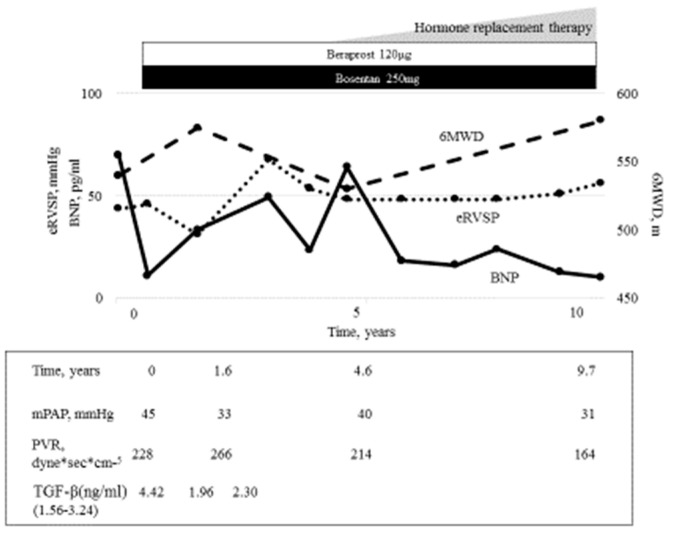
The clinical course most effective for a patient with PoPH. After the diagnosis of PoPH, therapy with bosentan and beraprost was started. The patient’s serum BNP level then decreased to within the normal limit. The patient’s 6MWD improved, and the PVR and mPAP decreased and were maintained at the improved level for 12 years. Serum TGF -β, which is one of most important factors for hepatic fibrosis, also decreased. RVSP, right ventricular systolic pressure; BNP, brain natriuretic peptide; PVR, pulmonary vascular resistance; mPAP, mean pulmonary arterial pressure; TGF-β, transforming Growth Factor-β; 6MWD, 6 min walk distance.

**Figure 4 jcm-12-00562-f004:**
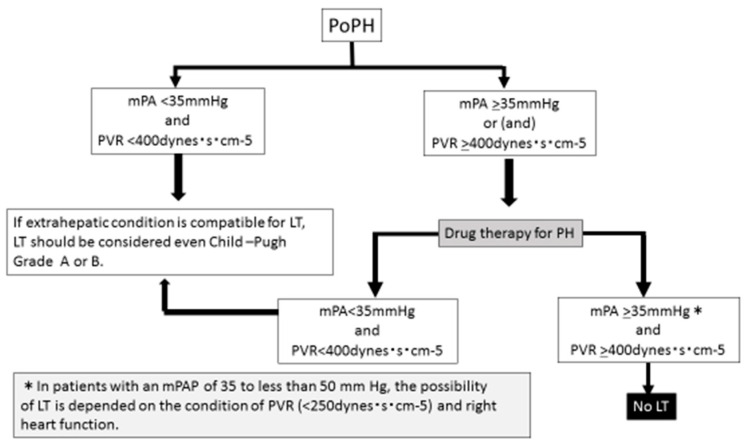
Flowchart of indications for LT in patients with PoPH. LT, liver transplantation; mPAP, mean pulmonary arterial pressure; PVR, pulmonary vascular resistance.

**Table 1 jcm-12-00562-t001:** Cases with portopulmonary hypertension.

No	Age Gender	Based Liver Disease	Shunt or Varices	BNP (pg/mL)	eRVSP (mmHg) mPAP (mmHg)	Therapy	Effect of Therapy	Prognosis
1	61 F	PBC	E, G-varices (E-transection)	45	144.6 -	No therapy		Dead
2	39 F	NASH-LC hypopituitarism	E-varices (EVL)	69.7	61 45	ERA, PRA	(+)	Survival
3	29 F	Portal vein obstruction (portsystemic shunt)	Huge shunt	52.2	59 34	ERA	(+)	Survival
4	60 F	PBC	E–varisce (EVL, EIS)	734	106 53	PDE-5i	(−)	Dead
5	82 F	LC-C	E-varices (E-transection)	124.9	55 -	No therapy		Dead
6	63 F	PBC HCV(+)	Huge shunt	69.7	122 45	ERA, PRA PDE-5i	(+)	Survival
7	20 F	Portal vein obstruction (portsystemic shunt)	Huge shunt E-varices	90.8	77 53	ERA, PDE-5i	(+)	Survival

F, female; PBC, primary biliary cholangitis; NASH, nonalcoholic steatohepatitis; LC, liver cirrhosis; E-varices, esophageal varices; G-varices, gastric varices, E-transection, esophageal transection for esophageal varices; EVL, endoscopic variceal ligation; EIS, endoscopic injection sclerotherapy; BNP, brain natriuretic peptide; eRVSP, estimate right ventricular systolic pressure; mPAP, mean pulmonary arterial pressure; ERA, endothelin receptor antagonists; PDE-5i, phosphodiesterase-5 inhibitor; PRA, prostacyclin receptor agonist.

**Table 2 jcm-12-00562-t002:** PoPH and HPS cases in our department (2000–2020).

	PoPH	HPS
Age (mean + S.E.)	50.6 ± 8.3	49.0 ± 5.9
Gender F/M (female %)	7/0 (100%) **	2/5 (29%)
Background liver disease	PBC 3 (43%) ^+^ LC due to HCV 1 (14%) ^+^ Portal-systemic shunt 2 (29%) NASH (14%)	LC due to HCV 4 (57%) NASH 2 (29%) Alcoholic LC 1(14%)
Shunt or varices	Huge portosytemic shunt 3 E, G varices 4 (surgical therapy 2, endoscopic therapy 2)	E-varices 3 Portosytemic shunt 1
Symptoms	Hearing failure 3 (43%) Dyspnea 3 (43%) No symptoms 1 (14%) <diagnosis during examination of liver tumor>	Dyspnea 7 (100%)
Prognosis and therapy	(1) Death due to liver and heart failure, 3 (43%) (2, no therapy; 1, drug therapy <not effective>) (2) Survival due to drug therapy, 4 (57%) <all effective>	(1) Death due to liver and respiratory failure, 3 (43%) (2) Survival, 4 (57%) (2, liver transplantation; 2, home oxygen and other therapy)

PBC, primary biliary cholangitis; HCV, hepatitis C virus; NASH, nonalcoholic steatohepatitis; LC, liver cirrhosis; E-varices, esophageal varices; G-varices, gastric varices; ** *p* < 0.01 + *p* < 0.1 by chi-square test.

## Data Availability

Data Availability Statements in section “MDPI Research Data Policies” at https://www.mdpi.com/ethics.

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
