# Peer review of "Current Therapy and Liver Transplantation for Portopulmonary Hypertension in Japan"

_jcm, 2023, doi:10.3390/jcm12020562_

Round 1

Reviewer 1 Report

The paper is presenting a review of current therapy options for portopulmonary hypertension in Japan, including liver transplantation.
The authors are presenting also a summary of the own centrum expertise on the management of portopulmonary hypertension and portopulmonary syndrome over twenty years (14 patients), which is consistent with the generally accepted approach. The review is in my opinion adequately covering the subject, current therapy options are being discussed pointing out the difficulties of using these
substances in patients with diminished liver function. PoPH is no longer seen as a contraindication to liver transplantation; criteria for allocating organs for these patients have been elaborated over the past two decades and PoPH constitutes a MELD exception.
The manuscript is in my opinion clear and well structured. The cited references are relevant to the topic. The figures are clear and easy to understand.
I find the conclusions are consistent with the argumentation; they are in concordance with previous published data and known challenges in the therapy of PoPH patients (altered liver function), including liver transplantation
(allocation criteria). The conclusions are supported by the listed citations.
The work is no novelty on the field but confirmatory, which is useful taking into
account the fact that few data was published about the characteristics of Asian
population with PoPH. On a PubMed search only one (the following) paper was
identified on this subject: Takahashi Y, Yamamoto K, Sakao S, Takeuchi T,
Suda R, Tanabe N, Tatsumi K. The clinical characteristics, treatment, and
survival of portopulmonary hypertension in Japan. BMC Pulm Med. 2021 Mar
16;21(1):89. doi: 10.1186/s12890-021-01452-3. PMID: 33726742; PMCID:
PMC7968246.
I suggest a minor revision of the language, with specific focus on the lines: 23-
24 pg.1; 118-126 pg.3

Author Response

Our point-by-point responses to the reviewers’ comments and questions are provided below.

#Reviewer 1

  1. The paper is presenting a review of current therapy options for portopulmonary hypertension in Japan, including liver transplantation. The work is no novelty on the field but confirmatory, which is useful taking into account the fact that few data was published about the characteristics of Asian population with PoPH. On a PubMed search only one (the following) paper was identified on this subject: Takahashi Y, et al. The clinical characteristics, treatment, and survival of portopulmonary hypertension in Japan. BMC Pulm Med. 2021 Mar 16;21(1):89. doi: 10.1186/s12890-021-01452-3. PMID: 33726742; PMCID: PMC7968246.

I suggest a minor revision of the language, with specific focus on the lines: 23-24 pg.1; 118-126 pg.3

→ Thank you for your positive review and useful suggestions. We added Takahashi’s paper (ref #30) in the references and added the following to the main text

“Takahashi et al. reported the clinical features and effect of PAH-specific drugs in Japanese patients with PoPH. Combined therapy showed good effect, and Japanese patients with PoPH showed higher cardiac outputs (COs) and cardiac indexes (CIs), better exercise tolerance, and lower PVRs than patients with idiopathic /heritable PAH.” in text. (Lines 185-189)

→ As suggested, we revised the language with specific focus on lines 23-24 and lines 118-126: “Previously, liver transplantation (LT) for PoPH was contraindicated; however, the indications for LT are changing and now take into account how well the PoPH is controlled by therapeutic drugs.”, and “Recently, lenvatinib and sorafenib have been frequently used as therapies for hepatocellular carcinoma (HCC) [22,23]. These are vascular endothelial growth factor (VEGF) targeting and/or tyrosine kinase inhibitor (TKI) drugs, and increase the levels of the vasoconstrictor endothelin, which binds to receptors on endothelial cells, causing smooth muscle contraction and increased vessel resistance and blood pressure. Ishikawa et al. published a case report of PoPH exacerbated by the administration of lenvatinib for HCC, suggesting that these drugs might worsen potential PoPH. When administering lenvatinib or sorafenib for HCC, the possibility of potential PoPH should be considered.” (Lines 139-146)

Reviewer 2 Report

Tokushige et al. reviewed the current knowledge of porpopulmonary hypertension (PoPH). Generally, this article was well-written and did provide an important insight to the easily-neglected disease in patients with cirrhosis. Some minor points should be refined to make perfect for this review.

1.     Figure 1: please show the full name of “HE” stain. Trichome should be shown in lower case.

2.     Line 71: general malaise and dyspnea due to pleural effusion. Actually, lots of factors including the presence of pleural effusion may result in general malaise and dyspnea, such as anemia, hepatopulmonary syndrome, hepatorenal syndrome and so on…

3.     Line 80: I agree with the screening echocardiography for cirrhotic patients. However, did the authors propose any prerequisite medical conditions (such as pulmonary symptoms that cannot be well- confirmed etc… ) that alert the physicians to arrange UCG to improve the positive predictive rate because the prevalence of PoPH is not high.

4.     Line 90: may be better shown in the term “hepatic venous pressure gradient (HVPG) > 5 mmHg”.

5.     Line 93: Tables 1 and 2 were missing. Please amend them. Was there any patients with coexistence of PoPH and HPS in your cohort?

6.     Lines 124-125: The sentence seemed hard to decipher and may be rewritten.

7.     Line 127: Dose the level of BNP increase in patients with HPS? I would be interested to know if there are divergent trends in PoPH and HPS, it would be simple to discriminate the two different clinical scenarios.

8.     Figure 2: The unit of BNP data before and after treatment should be removed since the vertical axis of BNP had shown the unit information. Furthermore, Case 6# BNP level after therapy should not be only 333 (should be higher than 633). Please confirm it.

9.     Figure 4: The footnote described “dependent on the condition of PVR”. Was there any cutoff value of PVR in such a condition?

Author Response

Our point-by-point responses to the reviewers’ comments and questions are provided below.

#Reviewer 2

  1. Figure 1: please show the full name of “HE” stain. Trichome should be shown in lower case.

→ We changed HE to “Hematoxylin-eosin stain.” “Masson-trichrome” has been corrected to use lower case.

  1. Line 71: general malaise and dyspnea due to pleural effusion. Actually, lots of factors including the presence of pleural effusion may result in general malaise and dyspnea, such as anemia, hepatopulmonary syndrome, hepatorenal syndrome and so on…

→ According to your suggestion, we changed “Actually, many factors, including the presence of pleural effusion, may result in general malaise and dyspnea, such as anemia, hepatopulmonary syndrome, hepatorenal syndrome, among others.” (Lines 72-75)

  1. Line 80: I agree with the screening echocardiography for cirrhotic patients. However, did the authors propose any prerequisite medical conditions (such as pulmonary symptoms that cannot be well- confirmed etc… ) that alert the physicians to arrange UCG to improve the positive predictive rate because the prevalence of PoPH is not high.

→ In line with your suggestion, we modified our text to read “However, this treatment option has certain problems in terms of cost and manpower given the low prevalence of PoPH. Echocardiography should be performed in selected LC patients, considering pulmonary symptoms, background liver diseases, and serum BNP.” (Lines 91-93)

  1. Line 90: may be better shown in the term “hepatic venous pressure gradient (HVPG) > 5 mmHg”.

→ We have changed this in the text in line with your suggestion - “hepatic venous pressure gradient (HVPG) > 5 mmHg”.

  1. Line 93: Tables 1 and 2 were missing. Please amend them. Was there any patients with coexistence of PoPH and HPS in your cohort?

→ We apologize for the missing tables. We added Tables 1 and 2. We experienced no patient with coexistence of PoPH and HPS in our cohort.

  1. Lines 124-125: The sentence seemed hard to decipher and may be rewritten.

→ We have rewritten these lines to improve readability, “When administering lenvatinib or sorafenib for HCC, the possibility of potential PoPH should be considered.” (Lines 145-146)

  1. Line 127: Dose the level of BNP increase in patients with HPS? I would be interested to know if there are divergent trends in PoPH and HPS, it would be simple to discriminate the two different clinical scenarios.

→ Out of 7 HPS patients, we measured serum BNP in 5 patients. In our data, BNP in HPSs was slightly increased, but the mean BNP of PoPH was significantly higher than that of HPS (mean serum BNP;40.6 ± 17.2 in HPS; 169 ± 250.3 in PoPH, <0.05). (Lines 151-154)

  1. Figure 2: The unit of BNP data before and after treatment should be removed since the vertical axis of BNP had shown the unit information. Furthermore, Case 6# BNP level after therapy should not be only 333 (should be higher than 633). Please confirm it.

→ The unit of BNP data before and after treatment were removed. We have corrected the typo for data “125”.

  1. Figure 4: The footnote described “dependent on the condition of PVR”. Was there any cutoff value of PVR in such a condition?

→ Krowka et al. reported that in patients with an mPAP of 35 to < 50 mmHg and a PVR of ≥ 250 dynes/s/cm, the mortality rate was 50%, with a PVR of < 250 dynes/s/cm it was 0%. Therefore, PVR is better < 250 dynes/s/cm, and we filled (<250 dynes/s/cm-5) in Figure 4.
